# A methodology and theoretical taxonomy for centrality measures: What are the best centrality indicators for student networks?

**Kristel Vignery**(ID)\*❦, **Wim Laurier**(ID)❦

Department of Economics & Management, Université Saint-Louis—Bruxelles, Brussels, Belgium

❦ These authors contributed equally to this work.
\* kristel.vignery@gmail.com, kristel.vignery@usaintlouis.be

**Data Availability Statement:** The data underlying this study are available at https://www.openicpsr.org/openicpsr/project/128601/version/V1/view (DOI: 10.3886/E128601V1).

## Abstract

In order to understand and represent the importance of nodes within networks better, most of the studies that investigate graphs compute the nodes' centrality within their network(s) of interest. In the literature, the most frequent measures used are degree, closeness and/or betweenness centrality, even if other measures might be valid candidates for representing the importance of nodes within networks. The main contribution of this paper is the development of a methodology that allows one to understand, compare and validate centrality indices when studying a particular network of interest. The proposed methodology integrates the following steps: choosing the centrality measures for the network of interest; developing a theoretical taxonomy of these measures; identifying, by means of Principal Component Analysis (PCA), latent dimensions of centrality within the network of interest; verifying the proposed taxonomy of centrality measures; and identifying the centrality measures that best represent the network of interest. Also, we applied the proposed methodology to an existing graph of interest, in our case a real friendship student network. We chose eighteen centrality measures that were developed in SNA and are available and computed in a specific library (CINNA), defined them thoroughly, and proposed a theoretical taxonomy of these eighteen measures. PCA showed the emergence of six latent dimensions of centrality within the student network and saturation of most of the centrality indices on the same categories as those proposed by the theoretical taxonomy. Additionally, the results suggest that indices other than the ones most frequently applied might be more relevant for research on friendship student networks. Finally, the integrated methodology that we propose can be applied to other centrality indices and/or other network types than student graphs.

## 1. Introduction

Many centrality measures for representing nodes within graphs exist in social network analysis (SNA). Many studies [e.g., 1–5] found correlations between several centrality indices within different network types. However, those researches did not deepened their results, i.e., did not explained their results in detail and /or classified the centrality indices based on the

**Funding:** The author(s) received no specific funding for this work.

**Competing interests:** The authors have declared that no competing interests exist.

correlations that were found. First, it is crucial to classify centrality measures, i.e., to create taxonomies of centrality indices, and to verify such theoretical classifications by means of thorough methodologies applied to real data. Second, in addition to the measures that are mostly used in network studies, alternative centrality indices could be considered when investigating such networks: since each centrality measure that has been developed in SNA differs in its meaning of centrality, using several measures in networks studies could bring more information about nodes centrality. Moreover, the appropriateness of the centrality measures should also be taken into account when studying networks, since each graph is different as far as its nature is concerned (e.g., its type: social, biological, financial, etc.; the direction of its edges: directed versus undirected).The paper's main contribution is to propose an integrated methodology that allows for choosing, comparing, and verifying centrality indices when investigating nodes within some network of interest. A subsequent contribution is the application of the proposed methodology to eighteen centrality indices computed from the CINNA library [6] and to an existing graph of interest: a friendship student network. To the best of our knowledge, the development of such a methodology, together with its application to real data, has never been done.

First, many centrality measures (e.g., degree, closeness, betweenness, eccentricity, geodesic $k$-path distance, eigenvector measure, Page rank score, etc.) were developed and defined in SNA to assess node centrality within a graph [7–14]. However, literature reviews on centrality measures are rare [15]. Furthermore, it is crucial to understand centrality measures in light of the network(s) of interest, i.e., in the context of the studied graph(s). Glossaries that thoroughly describe and define centrality measures within networks can serve this purpose. One of the objectives of this paper is to explain in detail a number of centrality measures found in the literature, that is, to give thorough definitions about those centrality indices, together with their formulas, in order to understand those indices in light of student networks, i.e., of our graph of interest.

Then, according to Lü et al. [15], it is important to compare and classify well-known centrality measures in order to highlight their similarities and differences. Theoretical, i.e., not yet empirically verified, classifications of centrality indices are useful to visualize [14] and understand those centrality measures better. As far as we know, such theoretical taxonomies of centrality measures [e.g., 12–15] are rare in the literature. Another purpose of the paper concerns the comparison and theoretical classification of centrality indices according to several criteria, such as their formulas, the benefits of high levels of centrality (e.g., access to information), and the consideration (or not) of neighborhood properties such as prestige [12–15]. Moreover, we found that the rare taxonomies of centrality measures that existed in the literature were not systematically verified, whether on real data or by thorough methodologies. However, in order to be sure to classify within same categories measures of centrality whose meaning is similar, and to separate within different categories centrality indices that are different in nature, it is crucial to verify the validity, accuracy, and appropriateness of the proposed classifications of centrality indices by means of criteria that are relevant for taxonomies. This paper therefore aims to develop a methodology that lets one verify theoretical taxonomies for centrality measures and that can be applied to any network type. An accurate taxonomy might also serve to validate classifications of centrality indices proposed in other studies.

Third, as stated before, many centrality measures exist in SNA, but according to our literature review, which is detailed further in the paper, it seems that only a few have been used. In studies related to the centrality of nodes within graphs, it is therefore useful to investigate other indices than those studied systematically. Also, the ways centrality is defined and computed might be relevant for best identifying central actors within graphs. In order to identify important nodes within a network, it is therefore crucial to test several centrality measures

based on the specific network type and the centrality definitions and to identify the most appropriate indices, i.e., that best represent the nodes within a network of interest among those measures [8–10].

We applied the proposed methodology to an existing graph of interest, in our case a real friendship student network. On the one hand, the question of selecting the best-suited centrality measures for representing a student network has received little attention in student networks research, and the literature on the topic mainly concerns other network types, e.g., biological networks, terrorist cells, and customer networks. Just as Ashtiani et al. [14] argue for biological networks, we argue that there is a need for guidelines pertaining to the relevance of centrality measures for student networks. Using relevant centrality indices might yield deeper understanding of friendship student networks and the mechanisms at work, e.g., the impact of student centrality within the peer group on the student's performance. On the other hand, applying the methodology that is developed in the paper to real data—our friendship student network- enables us to verify the validity of the proposed theoretical classification of centrality measures.

Our proposed methodology is composed of the following steps: (1) making the relevant choice for our network of interest, i.e., in our case a friendship student network, together with a thorough understanding and clear descriptions of some of the centrality measures that are computed in the CINNA library elaborated by Ashtiani & Jafari [6]; (2) developing and proposing a theoretical taxonomy of the chosen centrality indices according to several criteria, such as their definition, logic, and formulas; (3) identifying, by means of Principal Component Analysis (PCA), latent dimensions of centrality within our particular friendship student network; (4) verifying the proposed theoretical classification of centrality measures by comparing the theoretical taxonomy with the latent dimensions highlighted by the PCA and by means of useful criteria when validating taxonomies; and (5) identifying, also by means of the PCA results, the centrality measures that are the most representative and significant when identifying important nodes within friendship student networks. This research is exploratory and constitutes a first step to the development and applying of methodologies that compare and validate centrality measures which highlight central nodes within networks.

The four empirical research questions that are investigated in this paper are:

Research question 1) *Which centrality measures should be chosen among a large set of indices because they seem relevant for friendship student networks*?

Research question 2) *Which theoretical taxonomy might allow classifying those chosen indices*?

Research question 3) *By using PCA on those centrality measures, which are the centrality dimensions that are highlighted, and does the proposed theoretical taxonomy align with those dimensions*?

Research question 4) *Which are the representative centrality indices for friendship student networks*?

The second section of the paper relates to the theoretical background about (student) networks and centrality measures, together with their taxonomies for student networks. The third section presents the integrated methodology. The fourth section presents the data and friendship student network drawn from the data. The fifth section presents the results of the study: on the one hand, the comparison between the latent dimensions of centrality resulting from the PCA procedure and the theoretical classification of the centrality measures, and on the other hand, the centrality measures that contribute the most to representing the friendship

student network. Finally, the last section discusses the results and limitations of this study and points out the need for further research.

## 2. Theoretical background: Centrality measures and taxonomies for student networks

### 2.1. The relevance and use of centrality measures within (student) networks

A network is a set of nodes connected by edges or ties. For instance, with regard to student networks, a students' graph represents the students (i.e., the nodes) and their connections (i.e., the edges or ties) with other student(s) within the network. The literature recognizes the relevance of centrality measures for representing the importance of nodes in a graph [16]. The centrality concept gives information about its prestige, prominence, or involvement, how a node get access to and spreads information, and the node's proximity to phenomena that are observed within a network [16–19]. Various network studies in many fields have used centrality measures to represent nodes and possibly the links between the nodes' centrality and some variable(s) of interest. Among them we find studies in management and organizations [12, 20–26], economics and finance [27–31], marketing research [8, 32–35], sociology and political science [36–42], and biological networks [14, 43–51]. Also, student centrality within their network and the links with education outcome(s) have been the subjects of many studies. Those investigations concern performance and achievement [4, 24, 52–70], other aspects of learning (e.g., attitudes about the courses, sharing and construction of knowledge) [54, 71], delinquency [72, 73], sense of community [74, 75], and dropping out of school [76, 77]. Table 1 in S1 Appendix shows 63 studies conducted on centrality within networks (of which 27 were student networks), together with the centrality indices that were computed and used within those studies. In Tables 2 and 3 in S1 Appendix we computed the numbers and percentages of the centrality measures that were used within all network types (Table 2) or within student networks only (Table 3). Those percentages, which are represented in Fig 1, show that studies dedicated

**Table 1. Centrality measures taxonomy: Categories.**

|  | Topological structure of the network | | | Neighborhood | |
|---|---|---|---|---|---|
| Category number | 1 | 2 | 3 | 4 | 5 |
| Logic & formula | Geodesic distance-based | Geodesic path- based | Connectivity-based | Prestige of Neighborhood | Topology properties of neighborhood |
| Processes linked to centrality and advantages brought by high levels of centrality | (Speed of) access to information | Control over the information | Power and influence | Power and influence | Spread of Information |
|  |  | Spread of Information |  |  | Cohesiveness role |
|  |  | Bridge role |  |  |  |
| Eccentricity *(in- & out-)* | • |  |  |  |  |
| Closeness *(in- & out-)* | • |  |  |  |  |
| Residual closeness *(in- & out-)* | • | • |  |  |  |
| Geodesic *k*-path *(in- & out-)* | • | • |  |  |  |
| Betweenness |  | • |  |  |  |
| Bottleneck *(in- & out-)* |  | • |  |  |  |
| Eigenvector prestige score |  |  | • | • |  |
| Hub &authority scores |  |  | • | • |  |
| Page rank |  |  | • | • |  |
| Cross-clique connectivity |  |  | • |  | • |
| MNC *(in- & out-)* |  |  |  |  | • |

**Table 2. Structural properties of the respondents' network and of the augmented network.**

| | Respondents only | Augmented network |
|---|---|---|
| Number of nodes | 574 | 870 |
| Number of links | 1911 | 2550 |
| Smoothed number of links | 1260 | - |
| Reciprocity | 0.49 | 0.37 |
| Diameter | 27 | 18 |
| Average shortest path length | 9.40 | 7.13 |
| Degree distribution | KS.stat | KS.stat |
| | 0.07 | 0.05 |
| | *KS.p* | *KS.p* |
| | 0.68 | 0.93 |

to networks have mostly conceptualized centrality as (1) the simplest measure of centrality, i.e., degree centrality; (2) closeness centrality; and/or (3) betweenness centrality. However, as stated earlier, many other metrics have been developed in SNA in order to assess a node's centrality within a graph. S1 Appendix and Fig 1 show that (student) networks have rarely been represented using those alternative centrality measures.

Then, for network studies that concerned student centrality linked to some outcome of interest, some results showed centrality to have positive effects on education outcomes while others demonstrated no or even negative impacts. For instance, with regard to student performance, while the degree and closeness centralities seemed to have a positive effect on

**Table 3. Pearson correlations between the eighteen centrality measures for the augmented network.**

| | Ecc *in-* | Ecc *out-* | Clos *in-* | Clos *out-* | Res clos *in-* | Res clos *out-* | Be-tween | *k*-path *in-* | *k*-path *out-* | Bottle *in-* | Bottle *out-* | Eigen. | Page rank | Authority | Hub | MNC *in-* | MNC *out-* | Clique |
|---|---|---|---|---|---|---|---|---|---|---|---|---|---|---|---|---|---|---|
| Ecc *in-* | 1.00 | | | | | | | | | | | | | | | | | |
| Ecc *out-* | **0.17** | 1.00 | | | | | | | | | | | | | | | | |
| Clos *in-* | **0.96** | **0.18** | 1.00 | | | | | | | | | | | | | | | |
| Clos *out-* | **0.15** | **0.97** | **0.18** | 1.00 | | | | | | | | | | | | | | |
| Res clos *in-* | **0.44** | **0.23** | **0.63** | **0.26** | 1.00 | | | | | | | | | | | | | |
| Res clos *out-* | **0.13** | **0.59** | **0.19** | **0.72** | **0.36** | 1.00 | | | | | | | | | | | | |
| Between | **0.18** | **0.32** | **0.29** | **0.40** | **0.60** | **0.62** | 1.00 | | | | | | | | | | | |
| *k*-path *in-* | **0.33** | **0.21** | **0.51** | **0.24** | **0.97** | **0.35** | **0.62** | 1.00 | | | | | | | | | | |
| *k*-path *out-* | **0.12** | **0.46** | **0.17** | **0.60** | **0.33** | **0.97** | **0.62** | **0.34** | 1.00 | | | | | | | | | |
| Bottle *in-* | -.01 | -.04 | .00 | -.03 | -.02 | .02 | -.01 | -.02 | .03 | 1.00 | | | | | | | | |
| Bottle *out-* | 0.01 | -.02 | 0.01 | -.01 | 0.03 | 0.00 | 0.04 | 0.04 | -.01 | **0.21** | 1.00 | | | | | | | |
| Eigenvector | 0.00 | 0.02 | **0.08** | **0.06** | **0.21** | **0.10** | **0.09** | **0.21** | **0.10** | 0.00 | **0.06** | 1.00 | | | | | | |
| Page rank | **0.26** | -.07 | **0.35** | -.10 | **0.51** | -.03 | **0.26** | **0.49** | -.03 | 0.02 | **0.07** | 0.08 | 1.00 | | | | | |
| Authority | 0.01 | 0.02 | **0.08** | **0.07** | **0.20** | **0.12** | **0.11** | **0.21** | **0.13** | -.02 | 0.05 | **0.97** | **0.06** | 1.00 | | | | |
| Hub | 0.00 | 0.04 | **0.06** | **0.10** | **0.16** | **0.18** | **0.10** | **0.16** | **0.18** | 0.00 | **0.06** | **0.85** | **0.06** | **0.86** | 1.00 | | | |
| MNC *in-* | **0.27** | **0.21** | **0.35** | **0.25** | **0.50** | **0.33** | **0.32** | **0.48** | **0.31** | 0.03 | **0.08** | **0.35** | **0.30** | **0.37** | **0.31** | 1.00 | | |
| MNC *out-* | **0.13** | **0.37** | **0.20** | **0.41** | **0.32** | **0.48** | **0.24** | **0.31** | **0.44** | **0.10** | **0.08** | **0.32** | **0.24** | **0.34** | **0.43** | **0.68** | 1.00 | |
| Clique | **0.15** | **0.25** | **0.25** | **0.33** | **0.49** | **0.51** | **0.43** | **0.50** | **0.50** | 0.02 | **0.10** | **0.43** | **0.22** | **0.46** | **0.48** | **0.78** | **0.72** | 1.00 |

In bold front: coefficients for which the p-value is ≤ 0.05 (including those for which the p-value is ≤ 0.01).

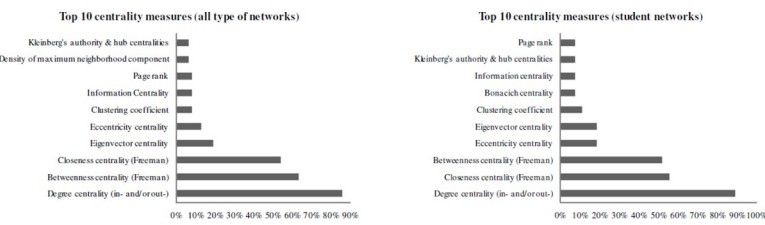

**Fig 1. Top 10 centrality measures in networks studies.**

achievement, the impact of betweenness centrality appeared to be less clear (see S2 Appendix for non-exhaustive instances of college student network studies and academic achievement). Together with other issues, such as the tie type (e.g., friendship versus strategic ties) that is investigated [4, 70], the choice of the centrality measures might explain the nature of the observed links between centrality and education outcomes, along with the inconsistencies in the findings [6]: The way centrality is computed mathematically defines centrality and establishes how individuals are represented within a network. Furthermore, studies showed that the ranks of the scores obtained for different centrality indices did not always match (e.g., a node could have high scores on some centrality measures, but average or low scores on other measures of centrality) [1, 8, 9]. As explained earlier, one objective of this paper is to highlight which centrality measures might be the most informative for friendship student networks. This research aims to select, from some chosen indices, the best-suited centrality measures for representing a friendship student network. Those centrality indices might then be used to study the impacts of student networks on education outcomes (e.g., student performance) in further research.

## 2.2. A theoretical taxonomy of centrality measures for student networks

Taxonomies have been defined as "*a formal specification of a shared conceptualisation*" [78]. Used in a variety of fields (pharmacology, engineering, physic, law, finance, etc.), they help to describe, organize, explain and predict phenomena; they yield knowledge about the relationships between different categories or objects; and they help researchers or practitioners to communicate about those phenomena [79–81]. Taxonomies may, however, "*be subject to a wide range of interpretations and misunderstandings*" [81]. Their appeal to a community (i.e., their sharedness) and fit with the reality they represent (i.e. their conceptualisation) therefore need to be validated [82]. Four criteria discussed in Guizzardi [83, 84] may be used in order to verify their fit with reality, namely, their soundness, completeness, lucidity, and laconicity (see Fig 2, where the constructs and/or objects colored in grey represent an absence of soundness, completeness, lucidity, or laconicity).

A taxonomy is *sound* when there is no construct excess and each of its constructs matches an underlying reality in an intended universe of discourse (e.g., within a student network). For example, a sound taxonomy of centrality measures for student networks contains only theoretical categories that represent centrality notions or dimensions for student networks in the real world faithfully. A *complete* (or exhaustive) taxonomy has no construct deficit and, hence, a construct for each aspect of the underlying reality [80]. For example, with an exhaustive list of theoretical categories, a complete taxonomy of centrality measures would reflect each centrality dimension existing within student networks, including those that have not been observed yet. A *lucid* taxonomy has no construct overload (homonymy) and hence only constructs that each maps to (at most) a single aspect of the underlying reality. For example, a lucid taxonomy would not contain a theoretical category that refers to several latent dimensions of student

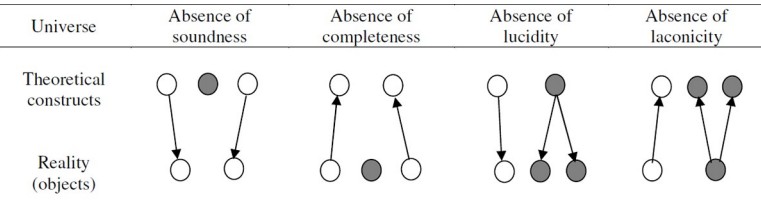

**Fig 2. Soundness, completeness, lucidity, and laconicity: Comparison between the theoretical classes or constructs and the reality or objects they represent.**

centrality. A *laconic* taxonomy has no construct redundancy (synonymy) and, hence, at most one construct for each aspect of the underlying reality. Classes must therefore be mutually exclusive, with no object that might belong to more than one category [80]. For example, a laconic taxonomy would contain only one theoretical category for each centrality dimension existing in student networks. In the absence of these criteria, the taxonomy could lead to ambiguous interpretations of the centrality measures pertaining to, for instance, a student network.

According to Lü et al. [15], "*a valuable work is to arrange well-known centralities and classify them.*" As stated before, numerous centrality indices exist in SNA, and clear theoretical classifications that are easy to use might help both experienced and novice researchers to understand and choose centrality indices from the vastness of options [32]. This work might be especially valuable for novice researchers in student networks, since taxonomies are useful to understand quickly the "*essential traits of the classified object by simply knowing in which category and with which other objects it has been grouped*" [80]. Moreover, centrality measures are related to the objectives linked to the use of those indices [9]. As Kozma et al. [79], who proposed a taxonomy of instructional treatments, different instructional treatment types having different impacts on learning and cognition, classifications of centrality indices might be valuable for efficiently visualizing and determining centrality measures that correspond to the goals of studies on student networks. For instance, if the purpose of one research initiative is to study the impact of a student's information control on some variable(s) of interest, such as learning or academic performance, taxonomies of centrality measures could help to select the most appropriate indices. Moreover, as stated before, the four criteria discussed in Guizzardi [83, 84] may be used to verify the validity of taxonomies that conceptualize the notion of centrality within (student networks). Now, not only has this work never been done before for student graphs, but a valid taxonomy of centrality measures tested on a student network might also be useful to generalize and validate the theoretical classifications of centrality indices proposed in other studies and for other graph types [e.g., in 12–15].

## 3. A proposed integrated methodology for studying centrality within networks

Fig 3 shows the integrated methodology. The first step consists of choosing a set of centrality indices according to not only the nature of the graph(s) considered (network type(s), an undirected versus a directed graph, etc.), but also the definition of the centrality indices, which must be evaluated in light of the specific network(s) of interest (social, biological, financial, etc.) and future research questions that will be investigated on the network(s). Second, a theoretical taxonomy of the chosen centrality measures is proposed. Third, those centrality indices are computed on one (or more) real network(s) of interest (in our paper, a single friendship student network). Fourth, PCA is applied to the computed centrality measures, and its outputs

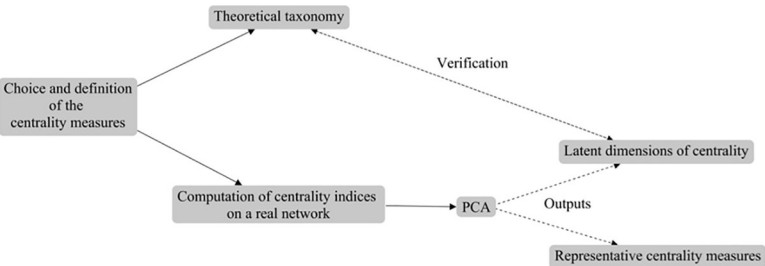

**Fig 3. Choosing, comparing, and verifying centrality indices on networks: An integrated methodology.**

are used (1) to verify the theoretical taxonomy of centrality indices by comparing this theoretical classification with factorial components emerging from the PCA, and (2) to determine the most informative centrality measures for the network(s) considered.

## 3.1. Choosing and defining the centrality measures

As stated before, our specific graph of interest is a friendship student network. We obtained a list of suitable centralities for our friendship student network by using the CINNA package [6] implemented in R©, and more precisely the function *proper_centralities*. The CINNA package can compute a great variety of centrality indices and is able to deal with directed and unweighted networks, which is the case for our graph. The function output, i.e., the complete list of the suitable centralities applicable to our graph, is presented in S3 Appendix. Among those suitable centralities, the measures for representing our friendship student network were selected by means of the sequence shown in Fig 4 and composed of the following steps:

1. *Who* are the nodes, *what* is/are the network type(s)? (i.e., in our case, a directed friendship student network; see the data section for the details);

2. A deep understanding of centrality measures is a necessary condition for (1) pursuing the methodology, (2) choosing the centrality indices from a vast set of measures, and (3) enabling deeper knowledge of the considered network(s) and the mechanisms occurring within the graph(s) of interest. With regard to the definitions of the centrality indices that were proposed by the CINNA library (see S3 Appendix for the proposed measures and S4 Appendix for the thorough definitions of the centrality indices that we chose), the second step consists of selecting which centrality measure might be suitable and interesting for further studies to investigate the links between some network(s) and outcome(s) of interest (in our case, between student centrality and education outcomes such as learning, performance, and so on). Our selection of the centrality indices—made in line with the perspective of a friendship student network—are justified below, when we present the centrality measures that we chose. Finally, for this second step, indices considered as irrelevant to the network(s) of interest and future research questions related to this/these network(s) were

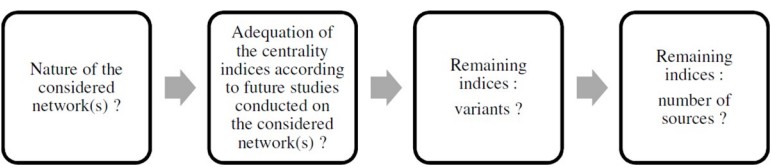

**Fig 4. Choosing networks' centrality indices: Sequence.**

not selected for further analyses. For instance, the current flow closeness centrality [85] is an index specific to electrical currents and was therefore not chosen since it is not suitable for social networks.

3. For the remaining indices: In the third step, we determined whether there were any measures with highly similar formulas, i.e., that differed by only a very few parameters. For instance, communicability betweenness centrality, flow betweenness centrality, load centrality, and stress centrality are all variants of betweenness centrality. We chose the centrality index figuring in the highest number of documents on Google Scholar and—as a benchmark—that was most used in network literature (see S1 Appendix): For betweenness centrality, we chose the measure that was proposed by Freeman [86].

4. For the remaining indices: In order to continue the methodological process with a reasonable number of indices, centrality measures that figured in very few documents on Google Scholar were not selected for further analyses.

Among the complete list of suitable centralities presented in S3 Appendix, the set of centrality measures chosen for our friendship student network is composed of eighteen centrality indices. Detailed definitions and explanations of these indices are given in S4 Appendix. We consider a centrality measure that takes into account the edge's direction, i.e., that can be computed separately on the incoming and outgoing ties, as two distinct indices. We explain, for each of the eighteen centrality measures, why they might be suitable for friendship student networks, and eventually for further studies conducted on those graph types and for their links with some education outcome(s):

*Eccentricity centrality* (computed separately on the incoming and outgoing ties): Eccentricity represents proximity versus isolation, i.e., the ease versus the difficulty of being reached by or to reach others within the network [66, 67]. As this centrality measure is related to the access of valuable information disseminated within a network [59], we found it important to investigate its representativeness for friendship student networks, since it was potentially linked to education outcomes of interest.

*Closeness centrality (Freeman)* (computed separately on the incoming and outgoing ties): This index was selected as a benchmark whose representativeness needed to be tested, since it is used mostly in the literature on student networks (see S1 Appendix). Furthermore, we included this measure because the closeness centrality concerns the speed or efficiency with which information will spread between nodes [12, 56, 87]: students with high levels of closeness centrality will enjoy efficient, easier, and faster access to information, advice, resources, and (academic) benefits in the network [12, 24, 26, 62, 64, 69, 88].

*Residual closeness centrality* (computed separately on the incoming and outgoing ties): Residual closeness centrality reflects the significance of a node as a communication link for its network [12], given that its removal significantly increases the distance between other nodes. This index was used to investigate nodes' centrality in several investigations [12, 14, 89]. However, as far as we know, no studies have yet used this measure on social networks, and we selected this index to assess its representativeness for such graph types, since being a strong communication link might be important in student networks.

*Betweenness centrality (Freeman)*: We also chose this measure as a benchmark to be assessed, since it has been used widely in previous studies on student networks (see S1 Appendix). Moreover, this index is interesting for studies conducted on student networks since

students with high levels of betweenness centrality connect other nodes and facilitate communication between other students in the network [26, 55, 56, 66, 67, 69]. Betweenness centrality represents the access and control that a node has over the (novel) information and resources contained in and flowing through a graph [9, 12, 15, 61, 69, 90], and therefore reflects its power or influence on other nodes [64, 87, 88].

*Geodesic k-path centrality* (computed separately on the incoming and the outgoing ties): This bounded *k*-betweenness index—proposed by Borgatti & Everett [91]- allows for computing the number of neighbors that are reachable by the fastest path up to length *k*, i.e., that are on a geodesic path less than *k* away. According to the authors, "*long*" shortest paths, which are considered when computing betweenness centrality, are not necessarily relevant for the spread of information through the network. Moreover, as stated before, the role of betweenness centrality in student network literature is still not clear. As far as we know, no research has yet studied geodesic*k*-path centrality in student networks, and we selected this index because it might be important for students by informing about their reception and/or dissemination of local information instead of the totality of information that circulates through the entire network [13].

*Bottleneck centrality* (computed separately on the incoming and outgoing ties): According to Obadi et al. [55], bottlenecks are "*central nodes that provide the only connection between different parts of a network*". Nodes with high bottleneck scores are therefore the most important ones in the network [92]. This index is used mostly in biological studies [44, 46, 50, 92, 93], but we decided to assess its importance for social graphs such as friendship student networks, since it represents the degree of confluence of links through a given node [93], i.e., through a given student in our case.

*Eigenvector centrality*: This centrality measures reflects the power, influence, or importance of a node in a network [8, 10, 61, 64]. The additional idea behind this score is that a node will be more prestigious or powerful if its neighbors are also central or well-connected [9, 18, 19, 26], an interesting centrality concept for social graphs. Few studies [e.g., 61, 64, 66, 67, 77] have investigated eigenvector centrality for student networks. We included this index since, given the advantages provided by high levels of eigenvector centrality, its representativeness for those graphs might be important and should therefore be assessed.

*Page rank*: The Page rank score [94, 95] quantifies the relative importance of a node within the network [61]. In social networks (e.g., friendship student networks), members that are cited by many individuals who have a high degree of Page rank will see their own Page rank increase [4, 19]. Only two [4, 61] of the twenty-seven studies concerning student networks that we reviewed used the Page rank score to investigate the centrality of students within their network, even though this measure might be a valid candidate for centrality and provides valuable information about the importance of a student within her/his friendship network.

*Hub & authority scores*: Related to networks, the authority score of a node reflects the importance of a node according to the number of important nodes, i.e., hubs that point towards it. Then, a node will be central if it points towards other important nodes, i.e., if it possesses a high hub score by pointing to good authorities. Here again, few studies [61, 77] have used those two distinct indices for investigating the central position of a student within her/his network, even though those measures provide different types of information, and might reflect specific and important centrality measures of a node within friendship student networks.

*MNC–Maximum Neighborhood Component* (computed separately on the incoming and outgoing ties): The MNC [46] is used mostly in the study of biological networks [e.g., 14, 47, 96, 97], even though it could also be used to identify central nodes in other graph types, such as human networks [46]. As far as we know, no study has already used the MNC to compute centrality within friendship student networks. We included this measure because the representativeness of this index could be interesting to estimate, since it concerns a student's centrality, which is related to the degrees of connectivity of her/his friends.

*Cross-clique connectivity*: The cross-clique connectivity [41] of a node represents the level of connectedness of this node to different sub-communities in a network. For a node, a high value of cross-clique connectivity represents its large influence in the graph, the spread and promotion of its ideas, its transfer of information between sub-communities in the network, and its role in the cohesiveness of its clique [13, 26, 87]. As far as we know, this way of representing centrality has yet not been the focus of research within friendship student networks, even though it could potentially highlight important information linked to (a student's) having a cohesiveness role within her/his network.

## 3.2. Proposed theoretical taxonomy

Based on the definitions of the eighteen indices that we chose (S4 Appendix) and examples of taxonomies from Song et al. [12], Lü et al. [15]; Ghazzali & Ouellet [13] and Ashtiani et al. [14], who each worked on different sets of centrality measures, we propose five theoretical categories to classify our eighteen centrality measures (see Table 1). To construct our theoretical taxonomy, we took three criteria that concern centrality measures into account, namely, (1) their formulas, (2) the benefits of high levels of centrality, and (3) the topological structure of the network and consideration (or non-consideration) of neighborhood properties. As the definitions of the eighteen indices show (see S4 Appendix), the first category (distance-based) assesses a node's proximity to the other members of the graph. The second category (geodesic path-based) is related to the geodesic paths on which nodes are located. The third category is based on connectivity, i.e., on the number of direct connections a node possesses. Finally, the fourth and five categories take the neighborhood of a node, the former neighbors' prestige, and, lastly, the topology of the members adjacent to the node in question into account.

## 3.3. Computation of the centrality measures on a real network

We used the igraph package [98] implemented in R© to compute the eighteen centrality measures for each student in our specific friendship student network.

## 3.4. Principal Component Analysis as a methodological tool

Previous studies have used various techniques to compare several centrality indices and/or highlight the most representative centrality measures within specific networks. Among those methods, we find, for instance, the computation of correlation coefficients between centrality indices [3, 13, 32]; Principal Component Analysis [6, 14, 99]; hierarchical clustering [e.g., 14]; the comparison between published network data sets and a Erdös–Renyi random network used as a baseline [10]; and more complex techniques, such as the influence maximization problem (IMP), heuristic and greedy algorithms, message passing theory, and percolation methods (see [15]).

According to Ashtiani & Jafari [6] and Ashtiani [100], PCA allows one to determine the most informative centrality indices and which centrality measures best represent the nodes within a network, i.e., which indices identify the central nodes most accurately.

PCA is a factorial analysis method that uses the correlations, i.e., inter-dependencies, between variables (in our case the eighteen centrality indices) to reduce the $p$-dimensional space of these variables to a $k$-dimensional space (with $k<p$). PCA results in a minimal number of principal components, i.e., factorial axis or latent dimensions that corresponds to maximum data dispersion, with these principal components being linear combinations of the initial variables. We performed PCA on our eighteen centrality indices and worked on standardized data to neutralize the problem of centrality measures with different units. We used the *Varimax* procedure as it ensures a better distribution of the variables over the factors by rotating the axis and allows easier interpretation of the factorial axis [101, 102]. We used SPSS 23 to perform PCA on our eighteen centrality measures.

First, we performed PCA on the centrality indices because the method can highlight the $k$ latent dimensions of centrality within a graph, in our case our friendship student network. Also, by computing the coordinates of each variable on each highlighted dimension, PCA enables one to identify the factor on which a variable has the highest loading, so that identifying the centrality indices that belong to the same latent dimension is possible in turn. Comparison of the PCA output—the $k$ highlighted latent dimensions—with the proposed theoretical classification of centrality indices then enabled us to check whether this theoretical classification could be validated.

Second, we performed PCA because it also enables one to determine the most representative centrality indices (among a complete set of measures) for a network of interest such as our student graph. The first step consists of retrieving the relative contribution of each centrality index (i.e., each $p$ variable) for each of the $k$ dimensions (i.e., the $k$ factorial axes) retained in the PCA. The following formula computes a variable's contribution to a factorial axis $k$:

$$Cont_{pk} = \frac{\rho(X_p, F_k)^2}{\sum \rho(X_p, F_k)^2} \tag{1}$$

Where $\rho(X_p, F_k)^2$ represents the quality of the variable's representation on the factorial axis $k$, and is equal to the squared correlation coefficient between the variable $X_p$ and the axis $F_k$; and $\sum\rho(X_p, F_k)^2$ represents the variance or inertia preserved on the factorial axis $k$, and is equal to the sum of the squared correlation coefficients between each variable $p$ and the factorial axis $k$ [102].

The second step then consists of computing each centrality measure's average contribution to the factorial plan, i.e., the average contribution on all $k$ factorial axes, i.e., $Cont_p$ We compared each average contribution to a threshold of $(1/18)\times100 = 5.55\%-$, i.e., in our case a centrality measure's theoretical contribution, since we worked on eighteen indices. For our paper, values higher or lower than 5.55% indicate a contribution that is above or below the theoretical average contribution, respectively.

## 4. Data

### 4.1. Collect of the data and management of the missing ties

The data were collected at Saint-Louis University in Brussels (i.e., USL-B), Belgium, in October 2016. The juridic department of the USL-B approved the study. The survey that was dispensed contained all the required information for an informed consent by the participants, and the survey was not mandatory. The data were anonymized before further analyses. Data collection took place during academic lectures that covered all curricula proposed by the university, i.e., law; economics; management sciences; literature, philosophy & history; communication, political science, & social science; and translation & interpretation studies. A total of 574 (43.95%

of the population) first-generation freshmen students (students registered in their first year of studies for the first time) completed a paper survey. They were asked about their friendship ties at university. In the survey, the student' friends were described as the "*persons with whom students spend personal time, with whom they interact on a regular basis (face to face, by phone, or on online social medias), whom they see outside classes, whom they trust, and/or with whom they share their personal issues*" [52, 55, 57].The nodes graph, i.e., the friendship student network, was drawn from the collected data. Since the survey was not mandatory, students who did not participate could nevertheless be cited as ties—a case of missing or non-respondent actors [103]. Thorough analysis of our graph revealed that 296 students were named as friends at least once by the 574 respondents but did not complete the survey and 651 of the 1911 designations (the total number of ties within the network) made by the respondents concern those missing students. Since SNA methods require complete recording of interactions between actors belonging to the studied network [16], we decided to impute the friendship relations for the 296 missing actors by means of Exponential Random Graph Models (ERGMs) [103] and statnet [104, 105], more specifically the ERGM package [106] implemented in R©. We tested progressively inclusive models and the final model used to simulate the ties on the missing actors included the effects of the number of edges in the network, node mixing by gender (gender was significant to predict the edge probabilities), node mixing by curriculum (the curriculum was significant to predict the edge probabilities), and a homophily effect for students in the same curriculum (i.e., nodes with the same curricula were more likely to be connected). All details (the justification for imputing the missing ties by means of ERGMs, the general formulas of the ERGMs, the terms used in our model, and its validation) are shown in S5 Appendix. The simulation enabled us to impute 639 ties to the 296 missing students. The total number of ties within the augmented network was therefore 2550. This imputation enabled us to compute centrality measures for each of the 870 students belonging to the augmented network, that is, the 574 respondents and the 296 missing actors to whom ties were imputed.

## 4.2. Comparison between the respondents' network and the augmented network: Visualization and structural properties of the two graphs

Since as explained earlier, SNA methods require the complete recording of interactions between actors within the studied network, in order to compare the augmented network to the respondents' network, we had to use—for this last network—the *complete cases* methodological approach, i.e., we had to delete within the respondents' network the nominations corresponding to students that did not complete the survey (i.e., we removed the 651 nominations made towards the missing actors).

Fig 5 shows the two networks: the original graph- i.e., the respondents' network—that has a number of nodes that is equal to 574, and the augmented network that has a number of nodes that is equal to 870. Table 2 shows the structural properties of the original network and of the augmented network.

The reciprocity (i.e., the percentage of dyads with mutual ties within a network: [18]) shows that, for the respondents, 49% of the links are reciprocal, and that, for the augmented network, the proportion of reciprocated ties is equal to 37%. The proportion of mutual ties is higher in the respondents' network probably because of the deletion of nominations towards missing actors, which has for effect to artificially increase the reciprocity. The diameter—i.e., the longest geodesic distance between any two students in the network [18, 61]—is equal to 27 for the network composed of the students who responded to the survey, and equal to 18 for the augmented network. Results show that the imputation of missing ties allowed decreasing the shortest distance between the two most distant students within the graph, which seems logical

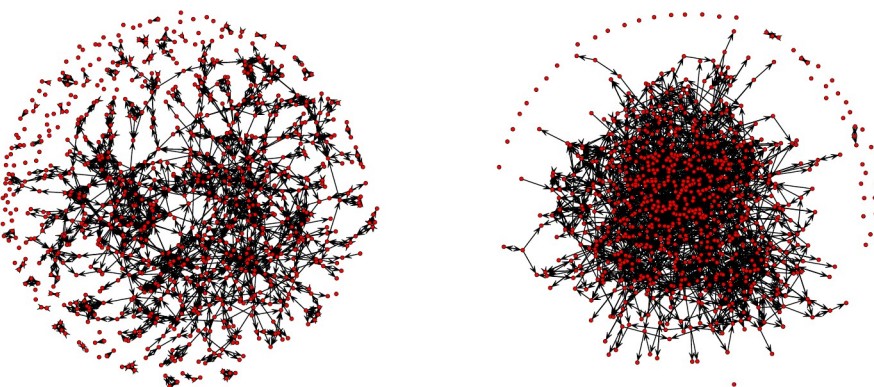

**Fig 5. Representation of the original network and of the augmented network.**

since in the respondents' network we deleted the nominations made by students towards missing actors. Then, the average shortest path length computes the mean of the geodesic distance between each pair of nodes within the network [107]. In average, the shortest path length between each dyad is equal to 9.40 in the respondents' graph and equal to 7.13 in the augmented network. The imputation process also allowed decreasing the average geodesic distance that is needed to access other students within the network. Finally, Fig 6 shows left-skew phenomena for the degree distribution of our two networks: the two histograms and cumulative density graphs show that there is a maximum of density at low values of node degree. Left-skewed degree distributions show similarity to scale-free networks [14]. In scale-free networks, most nodes have few links and only few nodes entertain many ties [108]. Since the probability of measuring a high value of node degree varies inversely as a power of that value [109], the distribution of nodes linkages in scale free-networks follows a power law [108]. The power law appears in many fields, including the social sciences [109]. As in Ashtiani et al. [14], we compared the degree distribution of both of our networks to the power law distribution in order to assess the scale-free property of our two graphs. Both of our networks seem to follow a power law distribution (Table 2). First, we observed small scores (which denote a better fit between the power law distribution and the data) for the Kolmogorov-Smirnov test statistic (i.e., resp. 0.07 and 0.08 for the respondents' network and for the augmented network). Second, if the resulting *p-value* of the Kolmogorov-Smirnov test is greater than 0.1, then the power law is a plausible hypothesis for fitting the distribution of nodes [110]. The two high *p-values* (i.e., resp. 0.68 and 0.93 for the respondents network and for the augmented network) show that the distribution of nodes for both our two networks do not significantly differ from the power law distribution (i.e., our two networks could have been drawn from the fitted power-law distribution). It is interesting to see that the augmented network seems to better fit the power-law distribution than the network composed only of respondents (again, probably because, for the respondents' network, we deleted the nominations that were sent towards students who did not answer to the survey).

## 5. Results

### 5.1. Correlations between the centrality measures

Three conditions are necessary for a PCA to be relevant. The first condition is that the variables must be correlated [111]. For the augmented network, the Pearson's correlation coefficients between each centrality measure together with their significance levels are shown in Table 3. The results show that the correlations between centrality measures that can be

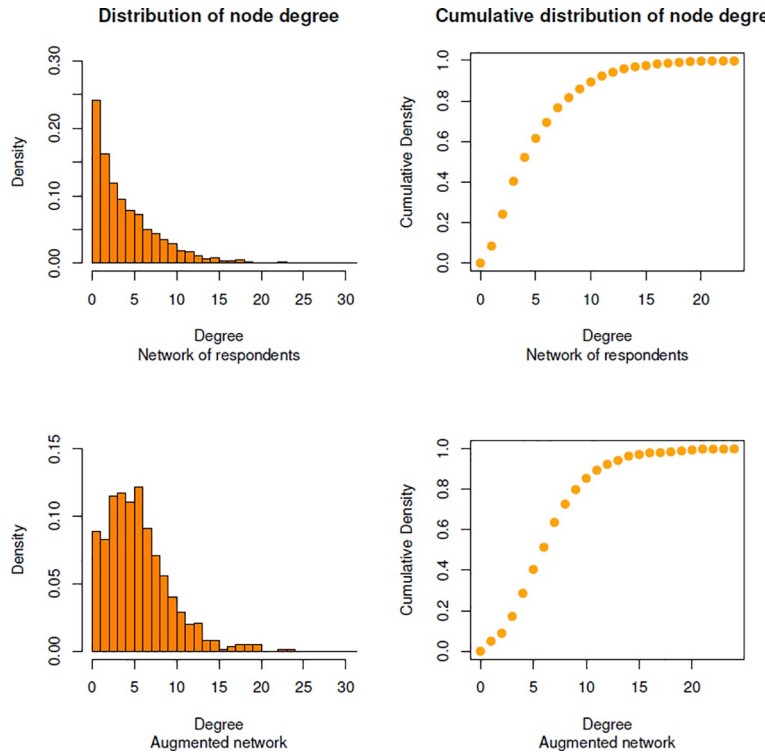

**Fig 6. Distribution of node degree for the two networks.**

computed separately on the incoming and outgoing ties (e.g., the eccentricity *in-* and *out-* centralities) are all positive and significant (*p-values* ≤ 0.05). Then, the results show many positive significant correlations between centrality measures (*p-values* ≤ 0.05), except for the bottleneck indices (*in-* and *out-*), for which few significant correlations are observed. We notice the strongest relationships between the eccentricity and closeness *in-* centralities (ρ = 0.96); the eccentricity and closeness *out-* centralities (ρ = 0.97); the residual closeness and geodesic *k*-path *in-* centralities (ρ = 0.97); the residual closeness and geodesic *k*-path *out-* centralities (ρ = 0.97); betweenness with the residual closeness *in-* (ρ = 0.62), geodesic *k*-path *in-* (ρ = 0.62), and geodesic *k*-path *out-* (ρ = 0.62) centralities; the eigenvector prestige and Kleinberg's authority centrality scores (ρ = 0.97); Kleinberg's hub centrality with Kleinberg's authority centrality (ρ = 0.86) and the eigenvector prestige scores (ρ = 0.85); the Page rank score with the residual closeness (ρ = 0.51) and geodesic *k*-path (ρ = 0.49) *in-* centralities; and cross-clique connectivity and the MNC: the maximum neighborhood component (*in-*: ρ = 0.78, and *out-*: ρ = 0.72).

## 5.2. The centrality latent dimensions within friendship student networks

For a PCA to be relevant, two other conditions (other than the correlations between variables) must be met. First, the Bartlett's test verifies whether highly correlated variables might be correlated to the same latent factor(s) [101, 111], which is the case ($\chi^2$ = 20512.07; *p-value* = 0.000). Second, the Kaiser-Meyer-Olkin index (KMO) tests the compressibility of information [111]. As the value of the *KMO* (= 0.713) is higher than 0.5 (the critical threshold), we can consider the factorization to be statistically acceptable.

Concerning the quality of a variable's representation, 50% of the information contained in each variable must be preserved in the factorial plan [101]. This fourth condition, which was computed automatically by SPSS 23, was met for all our centrality measures (see Table 4).

**Table 4. Sum of the squared correlation coefficients between a variable and each factorial axis.**

| Variables | Extraction |
|---|---|
| Ecc *in-* | 0.93 |
| Ecc *out-* | 0.78 |
| Clos *in-* | 0.96 |
| Clos *out-* | 0.88 |
| Res clos *in-* | 0.90 |
| Res clos *out-* | 0.89 |
| Between | 0.80 |
| *k*-path *in-* | 0.87 |
| *k*-path *out-* | 0.80 |
| Bottle *in-* | 0.62 |
| Bottle *out-* | 0.59 |
| Eigenvector | 0.95 |
| Page rank | 0.59 |
| Authority | 0.96 |
| Hub | 0.87 |
| MNC *in-* | 0.83 |
| MNC *out-* | 0.85 |
| Clique | 0.84 |

Then, to determine the minimum number of axes that preserves the maximum percentage of information, we used the Kaiser criterion, which consists of keeping the *k* factorial axes possessing eigenvalues (i.e. projected inertia/variance or information's degree preserved by an axis) that are higher than 1 [101, 111]. According to this criterion, six latent dimensions were retained. Moreover, they conserved 82.82% of the total inertia present in the original dimensional space (see Table 5). This respects an additional criterion based on a 60% threshold for the minimum percentage of conserved variance in the factorial plan [101, 111].

Table 6 shows the variables and their factor loading on the components for which their saturation is the highest, i.e., the latent dimensions on which the variables have the highest loading. The closeness, residual closeness, eccentricity, and geodesic *k*-path *out-* centralities are correlated with the first factorial axis. According to these four indices' definitions and formulas (in S4 Appendix), this first component or latent dimension might therefore reflect the ease with which a node reaches the other nodes, connects them, and transmits information throughout the network. The second dimension, which is highly correlated with Kleinberg's authority & hub centrality scores and the eigenvector prestige score, relates to centrality

**Table 5. Percentages of variance retained by the first six factorial axes.**

| Components | Initial Eigenvalues | | |
|---|---|---|---|
| | Total | % of Variance | Cumulative % |
| 1 | 6.07 | 33.71 | 33.71 |
| 2 | 2.80 | 15.54 | 49.25 |
| 3 | 2.48 | 13.77 | 63.02 |
| 4 | 1.32 | 7.33 | 70.35 |
| 5 | 1.23 | 6.82 | 77.17 |
| 6 | 1.02 | 5.65 | 82.82 |

**Table 6. Results of the Varimax rotation: Correlation of each variable on the factorial axis on which the saturation is the highest.**

| Centrality Indices | Components | | | | | |
| --- | --- | --- | --- | --- | --- | --- |
| | 1 | 2 | 3 | 4 | 5 | 6 |
| Closeness centrality (*out-*) | 0.91 | | | | | |
| Residual closeness centrality (*out-*) | 0.88 | | | | | |
| Eccentricity centrality (*out-*) | 0.83 | | | | | |
| Geodesic *k*-path centrality (*out-*) | 0.81 | | | | | |
| Kleinberg's authority centrality scores | | 0.96 | | | | |
| Eigenvector prestige score | | 0.96 | | | | |
| Kleinberg's hub centrality scores | | 0.91 | | | | |
| Geodesic *k*-path Centrality (*in-*) | | | 0.86 | | | |
| Residual closeness centrality (*in-*) | | | 0.83 | | | |
| Betweenness | | | 0.74 | | | |
| Page rank | | | 0.60 | | | |
| MNC—maximum neighborhood component (*out-*) | | | | 0.86 | | |
| MNC—maximum neighborhood component (*in-*) | | | | 0.82 | | |
| Cross-clique connectivity | | | | 0.72 | | |
| Eccentricity centrality (*in-*) | | | | | 0.94 | |
| Closeness centrality (*in-*) | | | | | 0.91 | |
| Bottleneck centrality (*in-*) | | | | | | 0.79 |
| Bottleneck centrality (*out-*) | | | | | | 0.76 |

through the number of connections with prestigious friends. The geodesic *k*-path and residual closeness *in-* centralities, betweenness, and Page rank score load on the third factor. This dimension might therefore denote the ability to control the received information and a node's degree of significance, especially by being located on the shortest (local) paths converging towards the node. The fourth dimension (the maximum neighborhood components and cross-clique connectivity) is linked to the degree of cross-connectivity of a student and her/his neighbors. The fifth component, which is highly correlated with the eccentricity and closeness *in-* centralities, relates to the ease with which a node is reached by the other nodes in the network and its ability to receive information. The sixth and last dimension reflects the degree of bottleneck, i.e., the degree of confluence through a given student.

### 5.3. Verifying the taxonomy on real data: A friendship student network

In order to verify the proposed taxonomy, we compared our theoretical classification (in Table 1) with the six centrality dimensions that emerged from the PCA that was performed on the augmented network, i.e., with the composition of the dimensions in terms of the eighteen centrality indices (in Table 6).

Within the taxonomy, four indices, namely, the eccentricity, closeness, residual closeness, and geodesic *k*-path centralities, are gathered within a first category (Category 1 in Table 1), which is built on the criteria of a geodesic distance-based formula and access to information as a centrality corollary. Those four indices, but computed for the *outgoing* ties only, saturate on the first latent dimension in the PCA (Table 6), which therefore matches Category 1 in Table 1. This theoretical category is therefore validated, but only for centralities computed on the nominations (declared friends) that are made by a node (student). Moreover, the closeness and eccentricity centralities that are computed on the *incoming* ties, i.e., the nominations received by a node, and which both saturate on the fifth factorial axis (Table 6), seem to form a subset within the first theoretical category in Table 1. Then, in the theoretical taxonomy, we

assigned the residual closeness and geodesic *k*-path centralities to the first category, but also to a second family of indices (Category 2 in Table 1) based on a geodesic-path formula and that relies on information control and diffusion. As seen above, residual closeness and geodesic *k*-path *out*- centralities have been verified to be part of Theoretical Category 1. However, they are validated for Theoretical Category 2 when they are computed on the incoming ties. They form a latent construct (the third dimension in Table 6) together with the betweenness index, with the latter also being validated for Theoretical Category 2 of centrality measures. As shown for the eccentricity and closeness centralities, the residual closeness and geodesic *k*-path indices therefore seem to be divided into two distinct categories according to the nature of the ties (i.e., *in*- versus *out*-),.

Then, the second latent dimension resulting from the PCA (Table 6), which brings together Kleinberg's authority & hub centrality scores along with the eigenvector prestige score, validates the third and fourth theoretical categories of indices within the taxonomy (Table 1), i.e., the categories based on the degree of connectivity and the prestige of the connections, respectively, which both reflect power and influence.

The fifth theoretical category in Table 1 concerns centrality indices that take the topology properties of the neighborhood into account in their formulas and relate to information spread and cohesiveness roles. The cross-clique connectivity and maximum neighborhood components (*in*- and *out*-) were proposed as being part of this category. That was confirmed by the PCA, which showed those three indices gathering on a same latent factor (the fourth dimension in Table 6). It should be noted that since a clique is composed of three or more nodes, cross-clique connectivity was also proposed as part of the third theoretical category in Table 1, which is based on the number of connections. However, the PCA confirmed that cross-clique connectivity belonged to the same category as the MNC scores.

Finally, bottleneck (*in*- and *out*-) and Page rank scores do not behave as expected according to the taxonomy. First, as shown above with the correlations and PCA, the two bottleneck indices do not correlate significantly with most of the other sixteen centrality measures, and both saturate on a specific factorial axis (the sixth dimension in Table 6). Yet based on the shortest paths in their algorithms, bottlenecks seem therefore to measure a different centrality type than the residual closeness, geodesic *k*-path, and betweenness indices. Second, we expected the Page rank score to be validated within the same theoretical category as the eigenvector and Kleinberg's authority & hub scores, since the Page rank formula takes the prestige of the incoming ties into account when computing a node's centrality. Instead, PCA showed a maximum saturation of Page rank on the same factorial axis as the geodesic *k*-path (*in*-), residual closeness (*in*-), and betweenness centralities.

## 5.4. Generalization and summary of the results

In order to generalize our findings, we computed the centrality indices on the original network (i.e., on the 574 respondents). Then, we applied the PCA to the eighteen centrality measures computed for those 574 respondents only and compared the PCA outputs with those from on the augmented sample (the 870 students). The tables related to the PCA performed on the 574 respondents are presented in S6 Appendix. The results show that even though some differences are highlighted between the two PCAs, we found roughly the same latent factors, which supports the generalizability of our results. The similarities and differences that were highlighted are as follows: (1) five factorial axes emerged for the PCA carried out on the 574 respondents instead of six latent dimensions for the augmented sample (see Table 5 in S6 Appendix); (2) the first dimension (which included the closeness *out*-, residual closeness *out*-, eccentricity *out*- and geodesic k-path *out*- centralities in the initial PCA) is similar between the

two PCA, except for the fact that betweenness is added to this dimension when the PCA is performed on the 574 respondents. However, as shown in Table 3 of the S6 Appendix, betweenness does not meet the requirement of 50% of the information preserved in the factorial plan. Therefore, its proximity to the four centrality measures that compose the first dimension should be considered with caution; (3) the second dimension (which includes the Kleinberg's authority, eigenvector, and Kleinberg's hub scores) matches with the dimension in the PCA carried out on the augmented sample exactly; (4) as in the initial PCA, the third dimension includes the geodesic k-path *in-* and residual closeness *in-* centralities, but, on the one hand, the eccentricity *in-* and closeness *in-* centralities are now part of this component, and, on the other hand, betweenness and the Page rank score are no longer included in this dimension. We stated earlier that betweenness is not well represented in the factorial plan, which, as for the initial PCA, is also the case for the Page rank score (see Table 4 in S6 Appendix). We also note that this dimension highlighted for the respondents only would validate the first category in our theoretical taxonomy, but only for incoming ties. This might be due to the fact that the incoming ties no longer include the imputed ties from the non-respondents; (5) the fourth dimension (which includes the MNC *in-* and *out-* together with cross-clique connectivity) corresponds to the latent factor generated by the first PCA, except for the Page rank score, which is now added to this dimension. But as just stated, its proximity to the three other centrality measures that compose this latent factor must be considered with caution; (6) in the correlation matrix (see Table 1 of S6 Appendix), the two bottleneck scores seem to be more linked to the other centrality measures than when computed for the 870 students. However, whether the PCA is performed on the respondents only, as on the augmented sample, they continue to form a unique latent dimension.

Table 7 compares the centrality dimensions that came out of the PCA (for both networks) with the theoretical categories proposed within the taxonomy, and summarizes the above findings: For both networks, the first latent dimension validates the first theoretical category, but for indices computed on the outgoing ties only, while for the augmented network (resp. the respondents' network), the fifth (resp. third) dimension matches, but only for incoming ties, two centrality measures that were proposed within Theoretical Category 1. For both networks, a unique dimension, i.e., Dimension 2, which includes and represents three indices, validates the membership in a unique class for those indices that were theoretically proposed as belonging to two theoretical categories, i.e., the third and fourth categories in Table 1. Then, for both networks, except for the Page rank score (for which we expected saturation on the second dimension) and the two bottleneck indices, which both saturate on the sixth (resp. fifth) dimension, the third dimension matches with Theoretical Category 2. However, for the respondents' network, the betweenness does not belong to the third dimension as it was expected. Finally, for both networks, the fifth theoretical category of centrality measures is validated by the fourth PCA dimension.

## 5.5. The most representative measures of centrality for friendship student networks

The last objective of the paper was to find the best centrality measures, i.e., the most representative and significant indices, when we investigate and represent friendship student networks. As in Ashtiani et al. [14] for biological networks, our goal was therefore to establish, from within a set of centrality indices, the measures that best categorize the central students and distinguish them from the peripheral ones.

As explained earlier in Section 3.4., we first retrieved the relative contribution of each of the eighteen centrality indices for each of the six dimensions that were retained in the PCA

**Table 7. The proposed theoretical classification and the centrality dimensions that came out of the PCA: Comparison.**

| Centrality measures | Theoretical constructs (From Table 1) | Latent dimensions (*reality*) for the augmented network (From Table 6) | Latent dimensions (*reality*) for the respondents' network (From Table 5 in S6 Appendix) |
|---|---|---|---|
| Eccentricity *(out-)* | 1 | 1 | 1 |
| Closeness *(out-)* | | | |
| Residual closeness *(out-)* | | | |
| Geodesic *k*-path *(out-)* | | | |
| Eccentricity *(in-)* | | 5 | 3 |
| Closeness *(in-)* | | | |
| Residual closeness *(in-)* | 2 | 3 | |
| Geodesic *k*-path *(in-)* | | | |
| Betweenness | | | 1 |
| Bottleneck *(in-)* | | 6 | 5 |
| Bottleneck *(out-)* | | | |
| Eigenvector prestige score | 3 or 4 | 2 | 2 |
| Hub score | | | |
| Authority score | | | |
| Page rank | | 3 | 4 |
| Cross-clique connectivity | 5 | 4 | |
| MNC *(in-)* | | | |
| MNC *(out-)* | | | |

As detailed above, the first category within Table 1 is based on geodesic distance and access to information, the second category is based on geodesic path and diffusion of information, the third category is based on connectivity and power, the fourth category is based on neighborhood prestige and power, and the fifth category is based on the neighborhood's topology and spread of information.

(performed on the augmented network). Then, we computed the average contribution of each of the eighteen centrality measures to the factorial plan (i.e., $Cont_p$). We compared each average contribution to the threshold of 5.55%, with higher (lower) values than this threshold indicating a contribution that is above (below) the theoretical average contribution. Table 8 shows the centrality measures and their average contributions (in descending order) to the factorial plan. The eight indices that best represent the variance of student centrality within a network are the bottleneck indices (*in-* &*out-*), eccentricity and closeness centralities computed on the incoming ties, maximum neighborhood component measures (*in-* &*out-*), and Kleinberg's authority & eigenvector prestige scores. On the contrary, other centralities (e.g., the betweenness index and eccentricity and closeness centralities computed on the outgoing ties) seem to contribute less to the factorial plan, being below the average threshold of 5.55%.

## 6. Discussion

We applied to a friendship student network the integrated methodology that we developed, i.e., (1) choosing, defining, and proposing a theoretical classification of centrality measures; (2) highlighting centrality dimensions within the network of interest; (3) verifying the proposed theoretical taxonomy by means of those dimensions; and (4) identifying representative centrality indices for friendship student networks.

**Table 8. Average contributions of the centralities to the factorial plan.**

|  | Average contribution to the factorial plan (Pct.) |
|---|---|
| Bottleneck (*in-*) | **8.50** |
| Bottleneck (*out-*) | **7.97** |
| Eccentricity (*in-*) | **7.95** |
| Closeness (*in-*) | **6.77** |
| MNC (*out-*) | **6.38** |
| MNC (*in-*) | **5.80** |
| Kleinberg's authority scores | **5.67** |
| Eigenvector prestige score | **5.64** |
| Betweenness | 5.53 |
| Kleinberg's hub scores | 4.94 |
| Closeness (*out-*) | 4.76 |
| Eccentricity (*out-*) | 4.63 |
| Geodesic *k*-path (*in-*) | 4.63 |
| Page rank | 4.22 |
| Residual closeness (*in-*) | 4.21 |
| Cross-clique connectivity | 4.11 |
| Residual closeness (*out-*) | 3.89 |
| Geodesic *k*-path (*out-*) | 3.69 |

In accordance with other studies conducted on other network types (e.g., sociometric networks in Valente et al. [3]; social, ecological, and neural networks in Batool & Niazi [10]; and terrorist and viral networks in Ghazzali & Ouellet [13]), our results show significant positive correlations between several centrality measures (e.g., between the eccentricity and closeness centralities, between the geodesic *k*-path centrality and betweenness index, and between the closeness centrality and eigenvector prestige score). Concerning the centrality dimensions that emerge in friendship student networks, our results show the existence of six latent constructs, namely, (1) a student's ability to reach friends and transfer information to them, (2) a student's ability to be reached by her/his friends and receive information from them, (3) a student's significance for the network structure and her/his control over the information flow, (4) a student's importance through the number of connections with prestigious students that are her/his friends, (5) the degree of cross-connectivity, and (6) the student's position as a confluent node. First, these results indicate that a centrality measure that is computed for incoming links seems to differ from the same centrality measure computed for outgoing links as regards its meaning and impacts on nodes. Our results show that the eccentricity, closeness, residual closeness, and geodesic *k*-path centralities that are computed for the outgoing ties saturate on a different latent construct than the eccentricity and closeness centralities that are computed for the incoming links. For friendship student networks, this result implies that it is not because a student is close to the other nodes of the network through her/his outgoing ties that s·he is automatically close to the other nodes of the network through her/his incoming connections. This also demonstrates that in friendship student networks, access to information might differ depending on a node's incoming and outgoing ties. Then, according to the nature of the ties (i.e., *in-* versus *out-*), the residual closeness and geodesic *k*-path centralities are also divided into two categories or dimensions. In other words, the number of friends that a student can reach—the student being located on (local) geodesic paths—might differ from the number of friends that can reach the student, also through (local) geodesic paths. Moreover, the fact that the residual closeness and geodesic *k*-path centralities are divided into two dimensions shows

that the outgoing links seem important for information access, while the incoming ties appear relevant for information control. In conclusion, a node might be highlighted as significant when centralities are computed on its incoming (outgoing) links, but not shown as central when its outgoing (incoming) ties are used in the computations. Second, PCA shows that Page rank saturates on the same factorial axis as the geodesic *k*-path (*in-*), residual closeness (*in-*), and betweenness centralities. As far as friendship student networks are concerned, these results suggest that students who are highlighted as central because they are cited by many other friends with high Page ranks might also be significant through the high number of neighbors that can reach them (i.e., those neighbors being located at maximum *k* steps towards them).From this we might infer that students with high Page rank scores are geographically close (on the graph) to the friend(s) who nominate them and belong to the same neighborhood as these students' closest friend(s). Finally, as stated before, the bottleneck measure seems to cover a particular centrality type that differs from those of the other indices that are related to location on geodesic paths within the network. According to the definitions of the centrality measures concerned (see S4 Appendix), this might be due to the fact that while the betweenness, geodesic *k*-path, and residual closeness centralities refer to the number of times a node is located on shortest paths between other nodes, a bottleneck provides the *only* connection between different parts of a network [56]. Consequently, for the betweenness, geodesic *k*-path, and residual closeness centralities, several students may be important by being located on shortest paths between other students, while students who are bottlenecks serve as the *only* bridges between several parts of the network, and therefore might be not only important but essential for the network. Future studies should be dedicated to a deeper understanding of the non-correlation between bottlenecks and other centrality measures that concern the locations of nodes on geodesic paths within student networks.

We matched the theoretical categories and the reality in order to verify whether the theoretical model could be validated.Except for the direction of the ties and/or for few indices, the five proposed categories of the theoretical classification correspond to the latent dimensions highlighted by the PCA:

1. The first theoretical category of the taxonomy is validated by the emergence of a first dimension, but for the centrality measures computed on the outgoing ties only.

2. Two centrality measures computed on incoming links that were proposed as belonging to the first theoretical category saturates on a second different dimension than the one that was expected.

3. Except for three indices, which saturate on two other dimensions than those that were expected, a third latent factor matches with the second theoretical category of the taxonomy.

4. Three indices that were theoretically proposed as belonging to two theoretical categories have the highest loading on a fourth unique dimension.

5. The last theoretical category of centrality measures is validated by the emergence of a unique dimension.

The results show that the integrated methodology applied to real data improved the taxonomy by adding some granularity. For instance, they highlighted that the direction of the ties should be considered in a theoretical classification of centrality measures. Then, regarding the four criteria (i.e., sound, complete, lucid, and laconic) that enable validating taxonomies, the methodology, when tested on real data, showed the following:

1. The proposed taxonomy seems to be *sound* (i.e., does not contain useless constructs), since each proposed theoretical category of indices matches with one latent dimension of centrality within the real network.

2. In order to be *complete* (i.e., to cover each aspect of the centrality notion within a friendship student network), an additional category within the theoretical taxonomy should be proposed for the bottleneck centrality.

3. In order to be to be *lucid* (i.e., contain categories that map to (at most) a single aspect of centrality), as explained above, categories that take the direction of the ties into account should be added to the taxonomy.

4. In order to be *laconic* (i.e., with no construct redundancy), the two categories "*Connectivity-based*" (Category 3 in Table 1) and "*Prestige of Neighborhood*" (Category 4 in Table 1) should be merged into a single construct, since they do not seem to relate to different aspects of centrality; that is, since the indices proposed for both categories—the eigenvector prestige score and Kleinberg's authority centrality scores—saturate on only one latent dimension. The page rank score would also needs to be reinterpreted for the taxonomy to be laconic, since this centrality measure was proposed as belonging to two theoretical categories, whilst being part of only one dimension in reality.

As stated before, applying our integrated methodology to real data could also be useful for comparing our verified theoretical classification with categories of centrality indices suggested by other authors. For the categories proposed by researchers from which we took inspiration to build our own taxonomy, we find several concordances, even if the classifications do not exactly match:

1. For Song et al. [12] we found a concordance with the distance and path-based categories for the closeness & betweenness centralities.

2. For Lü et al. [15] we found a concordance with their "*iterative refinement centralities*" category, which contains the eigenvector score and HITS (i.e., the Kleinberg's authority & hub centralities) algorithm.

3. For Ghazzali & Ouellet [13] we found a concordance with the closeness centrality, which is also included in a distance-based category; with the betweenness and geodesic *k*-path centralities, which are also part of a path-based category; and with the eigenvector score, which also belongs to a connectivity category.

4. For Ashtiani et al. [14] we found a concordance with the closeness, eccentricity, and residual closeness (but only *out-*) centralities that they proposed within a distance-based category; with the Kleinberg's authority & hub centralities included within a connectivity category; and with the maximum neighborhood component, which is also part of a neighborhood-based category.

Concerning the centrality measures that should be chosen to describe and represent friendship students within their network, as in Batool & Niazi [10] and in Ashtiani et al. [14], our results show centrality indices that contribute the most to the construction of the factorial plan and best reflect the variability of centrality within friendship student networks. For instance, future studies could use these seven measures that are rarely or not yet used in the literature on friendship student networks: (1) the two bottleneck indices, (2) the eccentricity *in-* centrality, (3) the two maximum neighborhood component measures, (4) Kleinberg's authority score, and (5) the eigenvector centrality measure. Using these indices could allow capturing and

investigating different dimensions of student centrality (i.e., its degree of confluence, ability to be reached and to receive information, degree of cross-connectivity, and centrality through prestigious connections), while making sure to select the best centrality candidates (reflecting a maximum of variability between students).

Four limitations of this research must be pointed out. First, the third and fourth steps of the sequence that allows choosing the centrality indices (in Fig 4) might be subjective. For the third step (the analysis of measures whose formulas are highly similar), measures that look alike regarding their formulas might—in some cases—behave differently from an empirical and/or mathematical point of view. For the fourth step, choosing not to perform further analyses on centrality measures that figured in very few documents on Google Scholar may be problematic, since it will contribute to *circular reasoning*: new measures might never get the chance to be tested. However, for this particular issue, the second step of the process provides the opportunity for choosing measures that were (very) rarely referenced in the literature (in our case: the residual closeness, bottleneck, and geodesic *k*-path centralities and the cross-clique connectivity).We proposed the idea of a sequence leading to a reasonable list of centrality indices, but future studies should continue investigating other measures (e.g., those whose formulas are similar to some extent or those that have rarely been referenced in the literature) according to the nature of their graph. Second, concerning PCA, each variable met the requirement of 50% of the information preserved in the factorial plan. But compared with the other centrality measures (for which the representation was greater than or equal to 78%), the Page rank score and bottleneck indices contained lower percentages of information, i.e., a maximum of 63%. If a variable is not well represented in the factorial plan, its correlation with another variable or other variables may be misinterpreted, i.e., variables might be considered close whereas that is not the case [102]. Therefore, the proximity of the Page rank score with the geodesic *k*-path centrality (*in-*), residual closeness centrality (*in-*) and betweenness should be considered with caution. The considered proximity between the two bottlenecks' scores should also be validated by subsequent studies. The third limitation relates to the high proportion (34%) of missing actors or non-respondents. Several authors [112, 113] have shown that high levels of survey non-response impact the structural properties of social networks and might cause underestimation of the computed coefficients [114]. Therefore, we chose the ERGM imputation technique (see S5 Appendix for the justification), in order to limit biases in the further analyses as much as possible. The comparison between the two PCA results (i.e., the results from the PCA performed on the respondents network and those from the PCA performed on the augmented network) show that choosing the ERGM imputation technique, and therefore the non-deletion of the nominations made towards the non-respondents, allowed our method to respect the three necessary conditions for the PCA to be robust (which was not the case when we performed the PCA on the 574 respondents, since betweenness did not meet the criterion of 50% of information conserved within the factorial plan), and increased the granularity of the PCA results by highlighting six instead of five dimensions. The fourth and final limitation relates to the distribution of the eighteen centrality measures, which are not normal distributions: Kolmogorov-Smirnov tests conducted on each eighteen centrality measures have rejected the normality of the distributions, the *p-values* being equal to 0.000 for each of the eighteen tests. Table 1 of S7 Appendix shows that some variables (e.g., the eigenvector, hub and authority scores) have a positively skewed and/or leptokurtic distribution, i.e., a high degree of positive skewness and/or of kurtosis. However, the Spearman correlations matrix (where the coefficient correlation are computed on the variables' ranks instead of the raw data) between the eighteen centrality measures shows that the pattern of correlations is similar than the one observed in Table 3, i.e., when using the Pearson's formula: Table 2 in S7 Appendix shows, on the one hand, significant correlations between most of the centrality

measures, and, on the other hand, the fact that the two bottleneck indices do not seem to be linked to the other centrality measures. Moreover, the total percentage of positive and significant correlations between all centrality measures except the two bottleneck indices is equal to 90.83% when using the Pearson's formula, and equal to 87.50% when using correlation on the ranks. Having used the raw data instead of their rank when we computed the correlations and performed the PCA seems therefore robust.

Finally, we see three prospects for future research.

1. First, the theoretical taxonomy could be tested on other student networks, but also in contexts other than academic settings, that is, on other network types (organizational, biological, etc.), in order to verify its validity (completeness, soundness, lucidity, and laconicity) and generalize its results. Also, to ensure its completeness by including centrality measures that have not been observed yet, other measures than those chosen in this paper and that can be computed in other libraries or web-based services (e.g., Hubba: hub objects analyzer from Lin et al. [46]; Centiserver from Jalili et al. [11]; Centilib from Gräßler et al. [85]) should be tested.

2. Second, our study should be replicated on other friendship student networks or other tie types within student networks, e.g., on strategic links, in order to validate the highlighted dimensions of centrality within student networks, but also to identify the best centrality measures when we investigate such graphs, since we worked on only one particular network. Batool & Niazi [10] have emphasized the need to pursue research that identifies the best centrality measures for a given network type (including the measures usually employed but also less traditional indices). Moreover, as stated by Landherr et al. [9], Batool & Niazi [10] and Ghazzali & Ouellet [13], more studies that compare and formalize centrality measures in different contexts are necessary. Centrality measures considered appropriate for a given network may not be able to identify central nodes correctly in other graph typologies [14]. For instance, it might be interesting to replicate this study on strategic ties within a student network, i.e., "*the people student would seek advice or assistance from and ask questions about their studies*" [53, 115], in order to investigate if the same dimensions of centrality and most representative centrality measures emerge. Moreover, we studied the centrality of students within a friendship network to get insights into the mechanisms occurring within those specific networks but also to orientate future studies that investigate the relationships between centrality and education outcomes such as student performance. As previous studies have found, centrality computed on different link types correlate differently with academic achievement (see, for instance, [4, 70]). Even if both network types (i.e., friendship and strategic links) are essential for education outcomes [115], this might be due to the fact that centrality within friendship versus strategic links procures different advantages for learning and performance [70], but also because different centrality types might be more or less representative and/or important according to the tie type considered.

3. The third perspective concerns frameworks or methodologies that could be used to verify the appropriateness of node centralities for different network types. Jalili et al. [11] identified 113 centrality measures; it would be very difficult, even impossible, to include all those measures in only one study. Along with Ashtiani et al. [14], we argue that the choice and identification of the best centrality candidates should be the first step when investigating networks to identify their key players. The integrated methodology that we propose might therefore be useful for future studies related to networks and node centralities (i.e., to test any set of centrality indices on any network and/or tie types).

## 7. Conclusion

In this research, we have proposed an integrated methodology that consists of: choosing—by means of thorough definitions and descriptions—a set of centrality indices, building a theoretical taxonomy of those centrality measures, highlighting latent centrality dimensions that exist within some network of interest, verifying the proposed taxonomy on real data by means of a robust statistical analysis (PCA), and pointing out which centrality measures should be used when investigating a network of interest. We applied our methodology to a friendship student network and we selected- in regards to our network of interest and to the centrality measures definition—relevant indices computed in the CINNA library (i.e., we investigated the *research question one*). First, the results demonstrate that for friendship student networks, the direction of the ties (incoming versus outgoing links) should be considered in the centrality computations, since they provide more information about a student's centrality within her/his peer network. Second, our results suggest that in the case of friendship student networks, six latent dimensions of centrality emerge for our eighteen indices, namely, the ability to reach friends and to transfer information; the ability to be reached and to receive information; the significance of a student for her/his network's structure, together with her/his control over the information flow; the student's importance through the number of connections with prestigious friends; the student's degree of cross-connectivity; and the student's position as a confluent node. Related to *research question three*, those six different latent dimensions should be integrated in future studies since they cover different aspects of centrality. Third, concerning the *research question four*, our results encourage using other indices, e.g., bottlenecks, eccentricity computed on the incoming links, the MNC measures, the Kleinberg's authority score, and the eigenvector measure, than those usually employed (e.g., betweenness centrality) when investigating friendship student networks. Fourth, in relation with the *research questions two and three*, the six latent dimensions that emerged from the PCA and the four criteria that make it possible to evaluate a theoretical classification, i.e., its soundness, completeness, lucidity, and laconicity, enabled us first to validate—for the most part—our taxonomy and, second, to compare our classification and find some similarities with categories of centrality proposed by other authors on other network types. Finally, the exploratory research methodology that we propose may constitute a first step when investigating some network(s) of interest, since it can be applied to other centrality indices (e.g., found in other libraries), other network types, and several tie types within graphs.

## Supporting information

**S1 Appendix.**
(DOCX)

**S2 Appendix.**
(DOCX)

**S3 Appendix.**
(DOCX)

**S4 Appendix.**
(DOCX)

**S5 Appendix.**
(DOCX)

**S6 Appendix.**
(DOCX)

**S7 Appendix.**
(DOCX)

**S1 File.**
(DOCX)

## Acknowledgments

This research was made possible with the assistance of University Saint-Louis–Brussels. The authors should like to thank five faculty members in particular (Philippe Desmette, Mauricio Garcia, Gilles Grandjean, Alexandre Girard & Laurent Van Eynde) for facilitating data collection. The authors also thank Professors François Fouss, Frédéric Nils & Marco Saerens, and the members of the MARAMI 2019 conference for their valuable rereading, comments, and suggestions. Finally, we should like to thank the two reviewers (i.e., Jesper Bruun and Peeters Ward) and Mohammed Saqr most warmly for their very constructive comments, which enabled us to improve the paper.

## Author Contributions

**Conceptualization:** Kristel Vignery, Wim Laurier.

**Formal analysis:** Kristel Vignery.

**Methodology:** Kristel Vignery.

**Resources:** Kristel Vignery.

**Supervision:** Wim Laurier.

**Validation:** Wim Laurier.

**Writing – original draft:** Kristel Vignery.

**Writing – review & editing:** Wim Laurier.

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
