## [Decision Letter · Decision Letter 0]

3 Apr 2020

PONE-D-20-06951

A methodology and a theoretical taxonomy for centrality measures: What are the best centrality indicators for student networks?

PLOS ONE

Dear Mrs. Vignery,

Thank you for submitting your manuscript to PLOS ONE. After careful consideration, we feel that it has merit but does not fully meet PLOS ONE’s publication criteria as it currently stands. Therefore, we invite you to submit a revised version of the manuscript that addresses the points raised during the review process.

 I agree with both reviewers that there are areas of improvements and points that you need to respond-to before considering this article further. Besides there are comments about the article I have also included.

We would appreciate receiving your revised manuscript by May 11 2020 11:59PM. To enhance the reproducibility of your results, we recommend that if applicable you deposit your laboratory protocols in protocols.io, where a protocol can be assigned its own identifier (DOI) such that it can be cited independently in the future. For instructions see: http://journals.plos.org/plosone/s/submission-guidelines#loc-laboratory-protocols

We look forward to receiving your revised manuscript.

Kind regards,

Mohammed Saqr, Ph.D

Academic Editor

PLOS ONE

Additional Editor Comments (if provided):

Section “2.1. Choosing and defining the centrality measures”

You have used the centrality measured offered by one R library, there are of course others offered by other libraries, in such an article where you are offering a taxonomy, you may need to be comprehensive, include - in your background or/in your analysis- other centrality measures that are commonly used and not offered by CINNA (Centiserver, keyplayer, igraph),

You probably need to start from a good review article and cover the relevant ones.This is an example.

Lü, L., Chen, D., Ren, X. L., Zhang, Q. M., Zhang, Y. C., & Zhou, T. (2016). Vital nodes identification in complex networks. Physics Reports, 650, 1–63. https://doi.org/10.1016/j.physrep.2016.06.007

I find the imputation step you have done to account for the missing values not fully explained, I am missing the ERGM formula, the terms used, the diagnostics and GOF. It might be good if you include a summary of these data in the text, and the full details in an appendix.

There have been concerns - I am not saying this is the case - that “the present submission is closely related to previously published and freely-accessible work including: Vignery Kristel and Wim Laurier. Achievement in student peer networks: A study of the selection process, peer effects and student centrality. International Journal of Educational Research 99 (2020): 101499”. I would like you to comment on the issue, specifically addressing that “that the separation into more than one article has not compromised the robustness of the statistical analysis (e.g. by removing required adjustments for multiple hypothesis testing).”

Journal Requirements:

2. Please proofread for numerical typos (commas instead of decimal points).

3. We noted in your submission details that a portion of your manuscript may have been presented or published elsewhere. [A part of the data that were used in this study was also used in an other study (already published) but for which the research questions and the objectives of the paper were totally different.] Please clarify whether this publication was peer-reviewed and formally published. If this work was previously peer-reviewed and published, in the cover letter please provide the reason that this work does not constitute dual publication and should be included in the current manuscript.

Reviewers' comments:

Reviewer's Responses to Questions

**Comments to the Author**

1. Is the manuscript technically sound, and do the data support the conclusions?

Reviewer #1: Yes

Reviewer #2: Partly

2. Has the statistical analysis been performed appropriately and rigorously? 

Reviewer #1: Yes

Reviewer #2: Yes

3. Have the authors made all data underlying the findings in their manuscript fully available?

Reviewer #1: Yes

Reviewer #2: No

4. Is the manuscript presented in an intelligible fashion and written in standard English?

Reviewer #1: Yes

Reviewer #2: No

5. Review Comments to the Author

Reviewer #1: The study has a lot of potential and the topic is relevant for the journal. The calculations are solid and support the taxonomy that is presented. While I would like to recommend this paper for publication, there are some issues and formal caveats that need to be resolved. In general, most of the sections in the introduction and literature review lack a clear rationale. More information is needed on data collection and analysis in Section III. Please have a look at my comments below.

ABSTRACT

The problem statement in the first two sentences of the abstract should be made more explicit. It is not clear that the way current SNA methods are applied is actually problematic. What does it mean for the field? And why is it relevant? (this information is mentioned later in the manuscript, but should be included in the abstract).

INTRODUCTION

While there is a rather clear problem statement presented in the introduction, the research rationale behind this study is missing. This needs to be included in the introduction. Consider rewriting this part. The literature review that follows must be separate from the introduction, and not be a sub-part of it.

1.1. The relevance of centrality measures in networks

The first part of this section is relevant. The second part is a list of applications, but without a clear purpose. Make sure that this list at least adds something to the research that is presented, i.e. make the links between the literature and your study more explicit.

1.2. The need for a taxonomy of centrality measures

In this section the research rationale is presented. Elaborate on the relevance of your work. Go beyond the fact that there hasn’t been done any research on this topic; rather explain why there should have been.

While the two research aims are presented in the different sections of the introduction (1. “develop a methodology that verifies the proposed theoretical taxonomy on real data”, 2. “selecting the best suited centrality measures for representing a student network or for studying the impacts of student networks on education outcomes such as student performance”), they are missing motivation and rationale. When re-writing this section, specify 1) problem statement, 2) motivation (what, how and why), 3) approach

In section 1.5. the approach on how to tackle the problem and achieving the goals is missing. This section generates more questions than it answers.

METHODOLOGY

Section III: Explain in more detail what type of data you used: mode (written, online, …) and way of collecting the data. Also mention the number of nodes and ties in the network. This is interesting for researchers who want to apply your findings to their own study.

RESULTS and DISCUSSION

Table 2. Specify: coefficients for which the p-value is ≤ 0. (seems to be incomplete)

On p. 19 “Yet based on shortest paths in their algorithm, bottlenecks seem therefore to measure a different type of centrality than the residual closeness, geodesic k-path and betweenness indices.” This requires an explanation.

This is repeated on p. 24: “In order to be complete (i.e., to cover each aspect of the centrality notion within a student network), an additional category within the theoretical taxonomy should be proposed for the Bottleneck centrality, which seems to cover a particular type of centrality.” As mentioned above, explain this ‘particular type of centrality’.

p. 22 “We applied the integrated methodology that we developed - i.e., (1) the choice, definition and proposal of a theoretical classification of centrality measures, (2) the highlighting of centrality dimensions within a network of interest, (3) the verification of the proposed theoretical taxonomy by means of those dimensions, and (4) the identification of representative centrality indices - on a real student network.” Yet, the steps you undertook are not elaborated upon extensively in the paper. Link this to your own data set. Detail the choices you made when analysing the student network in the methodology and results sections.

p. 25 other(s) variable(s) -> other variable(s)

Mind repetition in the limitations

CONCLUSION

Rework the conclusion. This section is a summary of the aims, but not of the work that has been done. Summarise the findings, rather than stating that you fulfilled the aims of the research study.

The same goes for the final sentence: state your main findings and possible recommendations.

In general, there are some language issues that need to be resolved. In the abstract alone there are a number of spelling mistakes (a nodes’ instead of a node’s), misplaced adverbs (e.g. ‘mostly’) and misuse of past and present tenses. In the body of the text, there are similar mistakes, as well as errors against subject-verb agreement (e.g. “how a node get access”) and typos (e.g. “the fourth and five categories”). These are just a couple of examples of the errors found throughout the manuscript. Please let the manuscript be proofread before resubmission.

Reviewer #2: First, I want to thank the authors for the opportunity to review their manuscript. I realise that a lot of hard work goes into manuscripts concerning student networks. I think the idea of trying to cluster network centrality measures is very good and needed, and I think that parts of the proposed methodology for doing is interesting and that the manuscript could become suitable for publication in Plus ONE. However, after carefully reading the manuscript, the appendix, and having browsed some of the references, I have a number of reservations and points of critique that I think needs to be addressed before publication. I believe that the manuscript should be reworked and perhaps some of the analyses redone (I’ll explain below). Therefore, I have opted for major revisions.

I have uploaded a commented document with this review, and authors should see this as what I think of while I read their manuscript, while I try to summarise here. As such, I would like to highlight:

1. Theory and concepts. I think the relevance of the use of ontology and taxonomy should be substantiated much more. Why are these relevant, and I particular, why is the theoretical taxonomy relevant? The reason I ask is because I do not see how the theoretical taxonomy explains network centrality measure in terms of e.g. student learning. So, if it is important that this is a *student* network, then the authors should spend more effort on connecting their student network to issues relating to student learning or student education in general. One thing that I would find important to theoretical considerations is the name generator and what a connection might mean. Since it is a friendship, any flow of information is inferred. Friendships can mean many things, and some friendships might not consist of sharing of information relevant to e.g. learning og education — even if the friendship is between two students. From the work I have participated in (Bruun & Brewe 2013, see link in comments in manuscript), the nature of connections was important to which centrality measures best predicted later grades. For example, Page rank was an important predictor for grades in a network pertaining to problem solving interactions, while Target entropy was the important predictor in a network pertaining to concept discussions. This was within the same cohort of students.

2. Methodological choices. Obviously, when the authors choose particular network centrality measures, as a reader, I will look for reasons why. I have not found convincing reasons for the first “proper centralities” nor for the 18 chosen centralities. Particularly since the authors claim that they have made a theoretical consideration based on the nature of the network, and I was not able to find these considerations in the text. Another methodological choice is that of PCA. Was this the choice made by the few others who have tried to cluster centrality measures? Why not perform a PCA on all the “proper” centrality measures? Did that violate the KMO-criterion of 0.5?

3. Generalisations of findings. I believe that it is very important that the authors are investigating a single network. This means that the taxonomy is not general but needs verification using other networks. It is mentioned in the end, but it should be clear from the beginning that this is a particular network with a particular name generator, particular students in a particular setting. The authors could expand a little on the generalisability by applying their methodology on the network without the ~300 non-responding students as well as the impute-connections network. If they find roughly the same rotated components, then this may be a good sign.

4. Target audience. From my perspective, it seems that the authors have a computer science background. I have a physics background, and other readers interested in student networks will have background from other science and research branches. Thus, we cannot be sure that relevant readers share our professional idiosyncrasies, and therefore, I think the authors should consider explaining concepts more. Throughout the manuscript, I have made notes where there are things I don’t understand completely, so authors may use this as an initial guide. But authors could also try to give their manuscript to a reader that they consider the target audience, but not in their own field of research for feedback.

I hope these considerations are helpful to the authors and would like to encourage their further work on this paper.

6. PLOS authors have the option to publish the peer review history of their article (what does this mean?). If published, this will include your full peer review and any attached files.

Reviewer #1: No

Reviewer #2: Yes: Jesper Bruun

---

## [Author Response · Author response to Decision Letter 0]

9 Sep 2020

The actions that were taken regarding the comments and recommendations made by the editor and the reviewers are described in the 'Response to Reviewers' file, and highlighted in the 'Revised Article with Changes Highlighted'. We would like to thank the reviewers and the editor most warmly for the very constructive comments that were made and enabled us to improve the paper.

---

## [Decision Letter · Decision Letter 1]

27 Oct 2020

PONE-D-20-06951R1

A methodology and a theoretical taxonomy for centrality measures: What are the best centrality indicators for student networks?

PLOS ONE

Dear Dr. Vignery,

Thank you for submitting your manuscript to PLOS ONE. After careful consideration, we feel that it has merit but does not fully meet PLOS ONE’s publication criteria as it currently stands. Therefore, we invite you to submit a revised version of the manuscript that addresses the points raised during the review process.

Revisers agree that the paper has improved substantially and many of their concerns have been addressed. Still, there are venues for improvement that the manuscript can benefit from.

We look forward to receiving your revised manuscript.

Kind regards,

Mohammed Saqr, Ph.D

Academic Editor

PLOS ONE

Reviewers' comments:

Reviewer's Responses to Questions

**Comments to the Author**

1. If the authors have adequately addressed your comments raised in a previous round of review and you feel that this manuscript is now acceptable for publication, you may indicate that here to bypass the “Comments to the Author” section, enter your conflict of interest statement in the “Confidential to Editor” section, and submit your "Accept" recommendation.

Reviewer #1: (No Response)

Reviewer #2: (No Response)

2. Is the manuscript technically sound, and do the data support the conclusions?

Reviewer #1: Yes

Reviewer #2: Yes

3. Has the statistical analysis been performed appropriately and rigorously? 

Reviewer #1: I Don't Know

Reviewer #2: Yes

4. Have the authors made all data underlying the findings in their manuscript fully available?

Reviewer #1: No

Reviewer #2: Yes

5. Is the manuscript presented in an intelligible fashion and written in standard English?

Reviewer #1: Yes

Reviewer #2: Yes

6. Review Comments to the Author

Reviewer #1: The paper has been substantially amended and both readability and transparency of the methods and data have benefitted from it. Most of my previous comments have been addressed and major issues have been resolved. The present study is an ambitious one and presenting such an extensive study in a limited number of pages might have resulted in a lack of depth in some areas. This comes to the foreground in the discussion section, where, in describing the features of the taxonomy, more suggestions for improvement are given than conclusive answers to the questions posed. Similarly, more than half of the space in this section is dedicated to suggestions for further research and to limitations. Maybe it would be beneficial for the credibility of this paper to highlight the exploratory nature of this study in the introduction. It has to be pointed out that, unless you would write a volume on the matter, most of the issues mentioned above cannot be resolved in the present form. With proper framing, this should not be an issue though.

Do mind some language errors such as “all types of network” in line 171 and “the second step consists in selecting” in line 291, among others.

Reviewer #2: I want to thank the authors for their substantial revisions to the manuscript. I believe that all comments have been addressed and mostly in a satisfactory way. However, a some issues remain or have been introduced.

Introduction

In essense, I believe that this section should convince the reader that the problem of finding out to what extent centrality measure measure the same "aspect of reality", is worthwhile. I believe that this is your main contribution (in concordance with what you write in lines 60-62. Here are some bumps on the way:

Line 55: "First, it is crucial to classify..." This sentence is a postulate that appears without backing (although I think the postulate is absolutely correcdt). Backing could come in the form of a reference, a further explanation or (preferably) both. One approach could be to point to other papers which have found correlations between centrality indices but have not treated these correlations any further. An example of such a paper is Bruun & Brewe (2013) - Figure 3, where different patterns of correlations are found for three different types of networks (with overlapping students). The authors could point to the shortcomings of that paper in this respect and build their case from there. Other papers might also be available to build a stronger case for this argument.

Line 57: The manuscript claims that closeness and betweenness centralities are most used. In fact, they have data to prove that, but the reader does not know this at this point. I suggest to leave out the specific mentioning of particular centrality measures.

Line 58: "Alternative measures could be considered..." Yes, but why would we want that? Explain this to the reader in a short phrase.

Line 59: ".. the appropriateness ... should also be taken into account.." Why? What would we gain from this?

Line 72: I do not see the logical imperative from the sparsity of reviews on centrality measures to thorough *definitions*. Rather, I would like to know, what centrality measures might mean in the context of student friendship networks; in the network of interest. In my opinion, the literature is abundant with rigorous definitions (eg. Wasserman & Faust 1994, Costa et al 2007).

Line 85: "rare taxonomies of centrality that existed..." Which - provide reference.

Line 87: "it is crucial to verify, ..." Why? Why will we potentially achieve by verifying these things? Better predictions? Better predicitve models? Other things?

Line 93: Only few centrality measure have been used... You can actually show this with your literature review. Perhaps rewrite this to say: "...*it seems* that only a few have been used.."

Line 96: How do we know that central actors have been identified correctly?

Also, either in the end of the introduction or in the end of Section 2, as a reader, I would very much appreciate clearly stated research questions in the form of:

"Research Question 1: Which theoretical and empirical taxonomies can be extracted using our PCA-based methodology for centrality measures?" <- my attempt...

Theoretical section

While Table 1 in Appendix 1 is informative, you could easily convert this into a graphical representation in the paper. Either a bar graph showing the ten most used measures or a word cloud. This is meant as a suggestion, only.

Figure 1 is very clear to me and should be kept.

Section III: A proposed ...

Line 489: For me it is really weird that this is not an equation. In my opinion, th quantity should be called some letter (eg. alpha), because then you can easily reference this quantity later on.

Line 490: Please write out the summation for \\lambda for clarity.

Section IV: Data

Here, I really miss an overview of the two networks (the "raw" as well as the imputed). I would like: number of nodes, N, number of links, L, diameter, average shortest path length, a summary of the degree distribution (like the power law exponent if they follow a powerlaw, or an average degree).

I would also prefer a graphical representation of each network to see how they look different. For example, seeing imputed links would be really good to see.

Also, in an appendix, it would be nice to have the distributions of the data (each of the 18 centrality measures). These can be easily computed in R. if their distributions differ a lot from normal distributions, this may influence the reader's view of the correlations.

Section V: Results

Since the authors have now made the calculations for the non-imputed network, I think they should include them throughout the paper (eg. in Table 6). This would increase the reader's understanding of the methodology (here, the impact of imputing links)

In Table 6, the authors could consider labeling Latent dimensions with "latent dimensions ("reality")" and "theoretical constructs" to make the connection to Figure 1 clear.

In Table 7, if these are averages, what are the uncertainties?

Section VI: Discussion

Given that I have understood Figure 1, here are some considerations:

Line 768: "are for the most part validated" - I do not think this wording is appropriate. I would start this section more "matter of factly": "We now turn to the question of whether the theoretical model could be validated." (or something like that). Then sum up which part of Figure 1 are validated and which are not (and how), as you do on the next page.

Lines 778-780: I am uncertain why the authors have placed the number 2 in the bottom right of Table 6, since they say here that a new category has to be proposed to achieve completeness. To make the point that the theoretical model is not complete that slot should be empty.

Lines 782-784: If you make 3+4 into one theoretical construct, would that not also point to two different empirical ("real") objects? Also, if the authors keep 2 in Table 6 then theres another theoretical concept which points to two empirical objects.

Lines 786-790: Here I would add that pagerank also needs to be reinterpreted for the mapping to be laconic, since empirical factor 3 would point to both theoretical categories 2 and 3+4. In the non-imputed network, as I understand it, pagerank is not associated with theoretical construct 2 either?

Section VII: Conclusion

Once the authors have the made research questions, they should also explicitly state what the answers are to the research questions here.

7. PLOS authors have the option to publish the peer review history of their article (what does this mean?). If published, this will include your full peer review and any attached files.

Reviewer #1: No

Reviewer #2: **Yes: **Jesper Bruun

---

## [Author Response · Author response to Decision Letter 1]

11 Nov 2020

The response to reviewers is in the document 'Response to Reviewers'

---

## [Decision Letter · Decision Letter 2]

4 Dec 2020

PONE-D-20-06951R2

A methodology and a theoretical taxonomy for centrality measures: What are the best centrality indicators for student networks?

PLOS ONE

Dear Dr. Vignery,

Thank you for submitting your manuscript to PLOS ONE. After careful consideration, we feel that it has merit but does not fully meet PLOS ONE’s publication criteria as it currently stands. Therefore, we invite you to submit a revised version of the manuscript that addresses the points raised during the review process. There are very few and small language issues that need to be fixed.

We look forward to receiving your revised manuscript.

Kind regards,

Mohammed Saqr, Ph.D

Academic Editor

PLOS ONE

Reviewers' comments:

Reviewer's Responses to Questions

**Comments to the Author**

1. If the authors have adequately addressed your comments raised in a previous round of review and you feel that this manuscript is now acceptable for publication, you may indicate that here to bypass the “Comments to the Author” section, enter your conflict of interest statement in the “Confidential to Editor” section, and submit your "Accept" recommendation.

Reviewer #1: All comments have been addressed

2. Is the manuscript technically sound, and do the data support the conclusions?

Reviewer #1: Yes

3. Has the statistical analysis been performed appropriately and rigorously? 

Reviewer #1: Yes

4. Have the authors made all data underlying the findings in their manuscript fully available?

Reviewer #1: Yes

5. Is the manuscript presented in an intelligible fashion and written in standard English?

Reviewer #1: Yes

6. Review Comments to the Author

Reviewer #1: Necessary changes have been made to the paper. I believe it has been beneficial for the credibility of this paper that the exploratory nature of the study has been mentioned in the introduction. With regard to language, the changes that have been made in green are not always correct.

- the fixed collocation with 'consist' is 'consist of'. An example would be: 'the second step consists of selecting'.

- 'type of information' should be 'types of information', 'other type of graphs' should be 'other graph types', 'other type of networks' should be 'other network types', etc.

Please resolve these before submitting the final version of the paper.

7. PLOS authors have the option to publish the peer review history of their article (what does this mean?). If published, this will include your full peer review and any attached files.

Reviewer #1: **Yes: **Ward Peeters

---

## [Editor Report · Decision Letter 3]

9 Dec 2020

A methodology and a theoretical taxonomy for centrality measures: What are the best centrality indicators for student networks?

PONE-D-20-06951R3

Dear Dr. Vignery,

We’re pleased to inform you that your manuscript has been judged scientifically suitable for publication and will be formally accepted for publication once it meets all outstanding technical requirements.

Kind regards,

Mohammed Saqr, Ph.D

Academic Editor

PLOS ONE
---

## [Editor Report · Acceptance letter]

16 Dec 2020

PONE-D-20-06951R3 

A methodology and theoretical taxonomy for centrality measures: What are the best centrality indicators for student networks? 

Dear Dr. Vignery:

I'm pleased to inform you that your manuscript has been deemed suitable for publication in PLOS ONE. Congratulations! Your manuscript is now with our production department. 

Kind regards, 

on behalf of

Dr. Mohammed Saqr 

Academic Editor

PLOS ONE